# Tissue Characteristics in Endodontic Regeneration: A Systematic Review

**DOI:** 10.3390/ijms231810534

**Published:** 2022-09-11

**Authors:** Sandra Minic, Sibylle Vital, Catherine Chaussain, Tchilalo Boukpessi, Francesca Mangione

**Affiliations:** 1URP 2496 Laboratory of Orofacial Pathologies, Imaging and Biotherapies, Life Imaging Platform (PIV), Laboratoire d’excellence INFLAMEX, UFR Odontology, Université Paris Cité, 92120 Montrouge, France; 2Louis Mourier Hospital, AP-HP, DMU ESPRIT, 92700 Colombes, France; 3Bretonneau Hospital Dental Department and Reference Center for Rare Diseases of Calcium and Phosphorus Metabolism, AP-HP, 75018 Paris, France; 4Pitié Salpétrière Hospital, DMU CHIR, AP-HP, 75013 Paris, France; 5Henri Mondor Hospital, AP-HP, 94000 Créteil, France

**Keywords:** regenerative endodontics, dentin-pulp complex regeneration, pulp injury, pulp necrosis, animal model

## Abstract

The regenerative endodontic procedure (REP) represents a treatment option for immature necrotic teeth with a periapical lesion. Currently, this therapy has a wide field of pre-clinical and clinical applications, but no standardization exists regarding successful criteria. Thus, by analysis of animal and human studies, the aim of this systematic review was to highlight the main characteristics of the tissue generated by REP. A customized search of PubMed, EMBASE, Scopus, and Web of Science databases from January 2000 to January 2022 was conducted. Seventy-five human and forty-nine animal studies were selected. In humans, the evaluation criteria were clinical 2D and 3D radiographic examinations. Most of the studies identified a successful REP with an asymptomatic tooth, apical lesion healing, and increased root thickness and length. In animals, histological and radiological criteria were considered. Newly formed tissues in the canals were fibrous, cementum, or bone-like tissues along the dentine walls depending on the area of the root. REP assured tooth development and viability. However, further studies are needed to identify procedures to successfully reproduce the physiological structure and function of the dentin–pulp complex.

## 1. Introduction

Tissue regeneration in dentistry has found clinical application in everyday practice with the regeneration endodontic procedure (REP). This treatment, relying on the triad of stem cells, scaffolds, and growth factors in a sterile environment (Figure 1), has the purpose of ideally replacing damaged structures, such as dentin and root, as well as cells of the pulp–dentin complex, in addition to the resolution of possible apical periodontitis [1]. In the case of pulp necrosis in immature teeth due to caries or developmental abnormalities, as well as dental trauma, it is recommended by several endodontic and pediatric dentistry associations to implement this therapy [2,3,4]. In fact, despite an important variability in the resolution of the clinical table, root edification, and neurogenesis, it is recognized as a viable management technique for immature permanent teeth with necrotic pulp [2,3,5], especially as it overcomes the challenges of conventional root canal treatment in the presence of short roots, thin fracture-prone dentine walls, and wide apices [6,7,8].

Indeed, REP has been a research topic for decades now, both in animals and in patients, with a wide range of protocols, materials, and success parameters evaluation. 

However, at the clinical stage, while high survival rates for REPs have been reported by numerous case reports, case series, and comparative clinical trials, the few existing systematic reviews highlight the weakness of these clinical trials [9,10]. Moreover, long-term follow-up prospective studies are necessary to better identify reliable success rates and outcomes of REP [11]. In fact, one of the major drawbacks is the level of accuracy in the assessment of regenerated tissues in REP-treated teeth [11], resulting in a lack of standardization regarding successful criteria. 

Thus, this systematic review aimed to highlight the main tissue characteristics related to the therapeutic success of REP in human and animal studies.

## 2. Methods

### 2.1. Search Strategy

The review process followed the Preferred Reporting Items for Systematic Reviews and Meta-Analyses (PRISMA) guidelines [12], and the protocol was registered in PROSPERO (International Prospective Register of Systematic Reviews) under the numbers CRD42022303001 for humans and CRD42022322610 for animals. 

In order to highlight the main characteristics of dental tissues treated by REP, based on the PICO strategy, this study systematically searched the following databases: PubMed/Embase/Scopus/Web of Science. The results were limited to the English language with a date range of January 2010 to February 2022, with full text available. The following combination of keywords was used: (Revitalisation OR Revitalization OR regenerative Endodontics OR revascularization OR regenerative Endodontics procedure OR regenerative Endodontics therapeutics) AND (dental pulp). 

### 2.2. Eligibility Criteria

Inclusion criteria were as follows: (1) All animal and human studies focusing on regenerative endodontic procedures; (2) orthotopic, semi-orthotopic, and ectopic procedures that attempted to revascularize or regenerate new pulp-like tissue; (3) English-language full text available; and (4) publication between 2010 and 2022. 

The exclusion criteria were as follows: (1) In vitro studies, (2) ex vivo studies, (3) in silico studies, (4) studies on transgenic animals, (5) publications solely in a non-English language, and (6) reviews and meta-analyses. The titles and abstracts were screened by two independent reviewers (S.M. and F.M.) against eligibility criteria. 

Full texts were analyzed whenever the abstract was not informative enough. A third reviewer (T.B.) was involved to resolve disagreements.

### 2.3. Data Extraction and Analysis

All selected articles were assigned depending on human or animal studies. Each article was subsequently classified according to (1) procedure, (2) follow-up, and (3) evaluation criteria.

### 2.4. Quality Analysis and Level of Evidence 

The risk of bias for animal studies was evaluated using the Systematic Review Centre Laboratory animal Experimentation (SYRCLE) risk of bias tool with the following criteria: (1) Selection bias, (2) performance bias, (3) detection bias, (4) attrition, and (5) reporting bias. Studies were scored with a ‘‘yes’’ for low risk of bias, ‘‘no’’ for high risk of bias, and ‘‘?’’for unclear risk of bias.

For human observational studies (cohort studies), the Newcastle–Ottawa Scale tool was used. Selections and outcomes were rated. Six binary responses contributed to an aggregate score corresponding to high risk (0–2 points), mild risk (3–4 points), and low risk of bias (5–6 points).

The risk of bias tool R.O.B 2.0 [13] was used to assess the quality of randomized clinical studies. Studies were scored with a ‘‘yes’’ for low risk of bias, ‘‘no’’ for high risk of bias, and ‘‘?’’ for some concerns of risk of bias.

ROBINS-I tools [14] were involved in assessing non-randomized clinical studies. By means of (1) randomization, (2) deviation from the intended intervention, (3) missing outcome data, (4) measurement of the outcome, and (5) selection of the reported results criteria, studies were scored with a ‘‘yes’’ for low risk of bias, ‘‘no’’ for high risk of bias, and ‘‘?’’ for medium risk of bias.

Finally, an adapted Newcastle–Ottawa Scale was used for case reports [15] with the following criteria: (1) Selection, (2) ascertainment, (3) causality, and (4) reporting. Studies were scored by eight binary responses and compiled into an aggregate score. The risk of bias was high (0–1 points), mild (2–3 points), or low (4–5 points).

The two reviewers analyzed all articles independently. Disagreement was resolved by discussion with TB and SV.

## 3. Results

### 3.1. Study Design and Characteristics of Included Studies

Details of the study selection process are outlined in Figure 2. The research retrieved 931 articles of which 774 were excluded at the title screening stage and 4 articles were excluded upon abstract screening. The full-text articles have been read in their entirety to see if they are relevant to our research. In total, 29 articles were excluded, and the reasons were as follows: 9 were not in vivo studies, 11 were on transgenic animals, and 9 studies focused on pulpotomy procedures.

Out of the 124 articles that met the inclusion criteria, 75 were clinical and 49 were animal studies. The results are summarized in Table 1, Table 2 and Table 3. All abbreviations are explained in abbreviation table. The details of each included study are provided in the Appendix A and Appendix A. 

### 3.2. Animals Studies

In this review, 49 animal models are represented. Small animals including rats, mice, rabbits, and ferrets were used for ectopic [16,17,18,19,20,21] and orthotopic REP models [22,23,24,25,26,27,28,29,30,31]. Large animals such as dogs, pigs, and sheep are used for orthotopic REP models [32,33,34,35,36,37,38,39,40,41,42,43,44,45,46,47,48,49,50,51,52,53,54,55,56,57,58,59,60,61,62,63,64]. 

#### 3.2.1. Ectopic REP

##### Procedure

Materials used for ectopic studies consisted of:
-Dentine slices or entire tooth roots: 

(1) With growth factors such as VEGF with or without cells as DPSCs [19,20].

(2) With cells in the presence of a collagen scaffold and calcium silicate cement [21]. 

(3) With Fibrin gel. [23].

-Polymers such as poly(lactic-co-glycolic acid) and rabbit DPSCs.

One study analyzed different disinfection agents in polyethylene tubes with triple antibiotic paste vs. calcium hydroxide paste [22] (Table 1, Appendix A).

##### Follow-up

Studies using ectopic procedures described follow-up periods from 12 days to 3 months [19,20,21,22,23,24] (Table 1, Appendix A).

##### Evaluation Criteria

Evaluation criteria were based on histological and immunohistochemical analyses. The goal was to analyze the type of tissue formed after implantation. Signs of inflammation (the presence of inflammatory cells) [19,22,24], signs of fibrosis [22,24], vascularization [20,21,24], and neurofilament [21] were evaluated. The nature of the new tissues that were synthetized inside the canal space was analyzed, and the presence or not of calcification was sought [21,23,24,25]. For IHC, different markers were used to characterize the pulp: DSPP or Nestin, vVW or CD34 for new vascularization, and PGP9,5 markers for nerve-like cell characterization (Table 1, Appendix A).

#### 3.2.2. Orthotopic REP

##### Procedure

Both small and large animals were used for REP. This therapy relied on the triad of tissue engineering: Stem cells, scaffolds, and growth factors, in a sterile environment. For that purpose, different materials were used: The gold standard MTA [26,27,28,29,30,31,32,33,34,35,36,37,38,39,40,41,42,43,44], different types of hydrogels [21,23,26,29,43,45,46,47,48,49], PRF or PRP alone or with cement or a blood clot (29,36,38,45,50–55,) or natural products, such as propolis [31,34,56]. Autologous pulp or cells such as DPSC or buccal fat were also used [20,21,32,50,51,57,58,59]. BMSCs with LPS vesicles or peptide angiogenic or dentinogenic or amelogenic [30,60] were also found. Growth factors such as VEGF were the most used [19,20,61]. (Table 2, Appendix A)

##### Follow-up

For small animal models, follow-up from 3 weeks to 12 weeks was achieved [22,23,24,25,26,27,28,29,30,31,65]. Regarding large animal models, the follow-up ranged from 1 to 28 weeks [28,32,33,34,35,36,37,38,39,40,41,42,43,44,45,46,47,48,49,50,51,52,53,54,55,56,57,58,59,61,62,63,64,66]. (Table 2, Appendix A).

##### Evaluation Criteria

To determine the success criteria, different techniques were used. Histology and radiology were dependent on the animal model used. Histological criteria consisted of periapical inflammation [34,36,37,39,42,46,62], inflammatory cell infiltration [32,34,35,37,38,39,43,48,56,61,66], the presence of pulp-like/vital tissue [33,34,35,36,37,40,41,42,43,44,45,46,48,49,51,52,53,55,56,57,62,63,64,66], the new formation of mineralized tissue [32,34,36,37,38,39,41,42,43,44,45,47,48,49,50,51,52,53,54,56,57,59,61,62,63,64,66], closure of the apex [22,24,25,34,36,38,41,42,44,47,48,49,51,52,57,62,66], the presence of odontoblastic palisade [33,35,36,42,45,46,54], and the presence of blood vessels [35,37,42,43,47,52,55,56,64], nerve fibers [35,42], or resorptions.

The criteria for the identification of different types of mineralized tissue were:-Dentin: Presence/absence of dentinal tubules.-Cementum: Absence of dentinal tubules and adherence onto dentin, and the presence of cementocyte-like cells.-Bone: Presence of Haversian canals with uniformly distributed osteocyte-like cells.-PDL: Presence of Sharpey’s fibers and fibers bridging cementum and bone.

For deeper characterization, IHC allowed specific protein localization. Specific markers such as DMP4, DLX1, GLI2, CEMP, CAP, NF, CD31, DSPP, SOX2, CGRP, and peripherin were identified in pigs, S100+ for neurofilaments, and DSSP, tenascin C, laminin, and fibronectin in dogs.

Bidimensional and/or tridimensional radiographies were performed to assess regeneration. The following radiological criteria were considered: Periapical radiolucency, root resorption, root thickening, root lengthening, and apex closure [22,23,26,27,31,32,34,41,42,43,46,49,51,57,59,63,64,66]. 

Based on histology and radiology, a scoring system was created to characterize the tissue and define the success of REP [23,37,38,48,49,50,52,53,61] (Table 2, Appendix A).

##### Others

qPCR was used to quantify the gene expression of different genes (DSPP, Col1A1, DMP1, and ALP) in order to determine whether tissue regeneration was triggered or not during revitalization [38]. (Table 2, Appendix A).

**Table 2 ijms-23-10534-t002:** Orthotopic REP techniques in animal studies.

Animal Models: Orthotopic REP Procedure				
Assessment	Main Results	Procedure	Follow-Up	Model
Histology	Presence of pulp-like / vital tissue	Gelatin and fibrin- based matrix BC alone + MTA PRP or PRF with cement and BC DPSCs and Buccal fat with BC and MTA. Nanosphere w/o BMSCs	3 months 3–7 months 3 months 1–2 months	Mini pig [34] Dogs [32,35,36,37,38,39,40,41,42,43,44,45,46,47,48,49,50,51,52,53,54,55,56,57,61,62,63] Ferrets [23,24,25,26] Rats [28,29,30,31]
	New formation of mineralized tissue	Gelatin and fibrine-based matrix BC alone + MTA PRP or PRF with cement and BC DPSCs and Buccal fat with BC and MTA.	3 months 3–7 months 3 months 3 months	Mini pig [34] Dogs [32,34,36,37,38,39,41,42,43,44,45,47,48,49,50,51,52,53,54,56,57,59,61,62,63,64,66] Sheep [64] Ferrets [23,24,25,26]
	Presence of odontoblastic palisade	Gelatin and fibrine-based matrix Autologous stem cells BC alone + MTA PRP or PRF with cement and BC DPSCs and Buccal fat with BC and MTA.	3–6 months 3–6 months	Mini pig [33,67] Dogs [33,35,36,42,45,46,54]
	Inflammatory cell infiltration	Gelatin and fibrine-based matrix Autologous stem cells BC alone + MTA PRP or PRF with cement and BC DPSCs and Buccal fat with BC and MTA. TAP + silver amalgam	3 months 3–6 months 3 months 1.5 months	Mini pig [34] Dogs [32,35,37,38,39,43,48,56,61,66] Ferrets [22,25] Rats [27]
	Presence of blood vessels	Gelatin and fibrine-based matrix Autologous stem cells BC alone + MTA PRP or PRF with cement and BC DPSCs and Buccal fat with BC and MTA.	3–6 months 3–7 months 3 months 1–1.5 months	Mini pig [33,34] Dogs [35,37,42,43,47,52,55,56,64] Ferrets [24] Rats [28,29]
	Presence of nerve fibers	SLan angiogenictarget peptide vs. SLed dentinogenic control peptide Autologous pulp + BC + MTA	3 months	Dogs [35,42]
	Presence of resorption	Gelatin and fibrine-based matrix Collagen sponge vs. PRF vs. MTA	3 months	Mini pig [34] Dogs [39]
	No intraradicular mineralized tissue deposition	Gelatin and fibrine-based matrix	3 months	Mini pig [34]
	Root maturation	BC + MTA	3 months	Sheep [64]
	Apex maturation	Gelatin and fibrine-based matrix Autologous stem cells BC alone + MTA PRP or PRF with cement and BC DPSCs and Buccal fat with BC and MTA.	3 months	Mini pig [34] Dogs [34,36,38,41,42,44,47,48,49,51,52,57,62,66] Sheep [64] Ferrets [22,24,25]
	Cementum cells/ tissue	Gelatin and fibrine-based matrix BC + Gelfoam BC + PRP BC + MTA	3–7 months	Mini pig [34] Dogs [41,48,50,52,56] Ferrets [24] Rats [28]
	Dentin tissue	BC + Gelfoam BC + PRP Propolis vs. MTA Autologous stem cells	1–3 months	Dogs [41,42,52] Rats [31]
	Osteodentin	(Buccal fat) vs. (BC + Buccal fat) + MTA BC + PRP BC + MTA	3–6 months	Dogs [38] Ferrets [26] Rats [31]
	Bone tissue	Autologous stem cells BC alone + MTA PRP or PRF with cement and BC DPSCs and Buccal fat with BC and MTA.	6 months	Dogs [39,48,49,50,52,53,59,60,63]
	Mineralized tissue deposition	Autologous stem cells BC alone + MTA PRP or PRF with cement and BC DPSCs and Buccal fat with BC and MTA.	3–6 months	Dogs [32,41,42,43,45,49,57,58] Ferrets [23,26]
Radiology	Presence of pulp-like / vital tissue	Gelatin and fibrine-based matrix BC alone + MTA PRP or PRF with cement and BC DPSCs and Buccal fat with BC and MTA. Nanosphere w/o BMSCs	3 months 3–7 months 3 months 1–2 months	Mini pig [34] Dogs [32,35,36,37,38,39,40,41,42,43,44,45,46,47,48,49,50,51,52,53,54,55,56,57,61,62,63] Ferrets [23,24,25,26] Rats [28,29,30,31]
	Apex closure	Autologous stem cells BC alone + MTA PRP or PRF with cement and BC DPSCs and Buccal fat with BC and MTA. TAP + silver amalgam	3–6 months	Dogs [32,41,42,43,46,49,51,57,58,59,66] Sheep [64] Ferrets [23,26] Rats [27]
	Increase root length	Autologous stem cells BC alone + MTA PRP or PRF with cement and BC DPSCs and Buccal fat with BC and MTA. TAP + silver amalgam	3–6 months	Dogs [32,41,42,46,49,51,57,58,59,66] Sheep [64] Ferrets [23,26] Rats [27]
	Increase dentin thickness	Autologous stem cells BC alone + MTA PRP or PRF with cement and BC DPSCs and Buccal fat with BC and MTA. TAP + silver amalgam	3–6 months	Dogs [32,41,42,46,49,51,57,58,59,66] Sheep [64] Ferrets [23,26] Rats [27]
	Periapical healing	Autologous stem cells BC alone + MTA PRP or PRF with cement and BC DPSCs and Buccal fat with BC and MTA. TAP + silver amalgam	3–6 months	Dogs [32,42,46,57,58,59] Sheep [64] Ferrets [23,26] Rats [27]
qPCR	DSPP, COL1A1, ALP, DMP1 expression	(Buccal fat) vs. (BC + Buccal fat) + MTA	3 months	Dogs [38]

### 3.3. Human Studies

#### 3.3.1. REP protocol

All human studies consisted of randomized, non-randomized, case reports, and retrospective studies. The evaluation criteria were clinical examination, 2D and 3D radiography, and/or MRI. 

Many protocols have been tested to achieve pulp-like tissue regeneration. Regarding the cement used, among all included studies, MTA was the most frequently used cement [7,8,60,67,68,69,70,71,72,73,74,75,76,77,78,79,80,81,82,83,84,85,86,87,88,89,90,91,92,93,94,95,96,97,98,99,100,101,102,103,104,105,106,107,108,109,110,111,112,113]. Seven studies were performed with Biodentine [75,85,114,115,116,117,118], one study used a Calcium-Enriched Mixture [119], two studies used Synoss Putty [120,121], one used calcium hydroxide [6,90], and five used Glass Ionomer Cement [79,112,122,123,124] to create a mineralized bridge to close the pulp chamber. Different types of materials were used to regenerate pulp tissue. Most of the studies only used blood clots as a scaffold. Some used other supplemental scaffolding components such as collagen [72,74,75,76,81,83,88,90,93,115,117,124,125], PRF or iPRF (+ MTA [8,87,91,126]; + Biodentine [116,117,118] + BC [87,91] + Portland cement [124]); and PRP [7,65,69,71,89,91,95,105,122,126] in addition to the calcium silicate cements. Blood clots have, at times, been used alone as a control [65,91,121,122,127] in comparative studies. Two studies have used collagen scaffolds with two types of cells, such as MDSPCs or umbilical cells MSCs [115,123].

Four studies have investigated the best canal disinfection technique in use and identified bi-antibiotics and triple-antibiotics of calcium hydroxide [128,129,130,131]. However, it is also important to identify whether the procedure of REP is more successful when performed in one visit or two [132] (Table 3, Appendix A).

#### 3.3.2. Follow-Up

According to the studies, different follow-up timepoints were reported. For retrospective studies, follow-up periods of 1 month to 8.2 years were observed [73,74,75,76,77,78,79,80,81,82,83,84,85,99,126,132,133,134]; randomized studies employed follow-up periods of 21 days to 18 months [6,7,65,69,70,71,85,86,87,88,89,90,91,114,115,121,126,128,129,134]; non-randomized studies used periods of 2 weeks to 36 months [66,116,121,123,124,132,135]; and cases series employed periods of 1 week to 6 years [8,67,92,93,94,95,96,97,99,100,101,102,103,104,105,106,107,108,109,110,111,112,113,117,118,119,120,127,130,136] (Table 3, Appendix A).

#### 3.3.3. Clinical and Radiographic Evaluation of REP

For clinical examination, teeth could be symptomatic or not with different sensitivity tests (thermal or electric), percussion, pain on percussion, and palpation. Dyschromia, swelling, and tenderness of the surrounding tissues were also assessed. The mobility of teeth, as well as pocket probing, were evaluated.

Radiographically, the teeth should have shown the resolution of the apical lesion with PAI scoring, apical closure, root length growth, thickening of the root walls, and the formation of calcific barrier. On tridimensional radiographic observation, the lesion size, bone density, root length, and pulp area were evaluated. Rarely, MRI was used in order to identify more organized tissue in the canal, dentin deposits, or mineralization (Table 3, Appendix A).

#### 3.3.4. Others

Other techniques such as qPCR were used to quantify how many bacteria were present on the canal dentin walls after different interappointment medications [128,129] and identify stem cells in the canal [135]. Histology was used for analysis in cases of a crown fracture [107,127] or tooth extraction as a result of orthodontic reasons [106,107,121,136] (Table 3, Appendix A).

**Table 3 ijms-23-10534-t003:** Human studies of REP.

Human Model: Regenerative Endodontic Procedure				
Assessment	Main Results	Procedure	Follow-Up	Articles
Clinical tests	Asymptomatic teeth	BC + Biodentine or MTA BC +PRF + MTA or Biodentine or GIC BC + PRP + MTA or Biodentine or GIC BC + Collagen + MTA or BC+ UC-MSCs + collagen + MTA BC + PRF + Collagen + Biodentine or Portland mDPSCs + G-CSF + Collagen + MTA Medication on different Appointment TAP vs. CaOH_2_ vs. formocresol Bi antibiotic + GIC	21 days–79 months	[61,66,68,69,70,71,72,86,87,90,91,94,95,96,97,101,102,103,104,106,107,110,111,113,116,117,118,119,120,121,122,123,124,125,126,128,130,131,132]
	PAI	BC + MTA Sealbio vs. obturation	12–24 months	[80,98,134]
	Dyschromia	BC + collaplug MTA vs. Biodentine vs. GIC BC + MTA vs. Biodentine BC + PRF vs. PRP + MTA Bi-antibiotic paste + BC + GIC	12–96 months	[75,79,85,86,98,126,130]
	Mobility	BC + Synoss Putty	72 months	[120]
Radiographic observation	Apical lesion	BC + hydrogel with FGF+ MTA BC + DPSC In hydrogel + MTA or GIC BC + MTA BC + PRF + Biodentine BC + PRP + MTA BC + PRF vs. PRP + Collagen + GIC BC + Synoss putty + MTA BC + Collagen + Portland + MTA BC + LPRF + Portland cement	21 days– 72 months	[8,66,70,71,79,80,81,82,83,90,92,93,95,97,99,100,103,105,107,112,113,116,118,120,122,124,125,128,133]
	Root length	BC + hydrogel with FGF+ MTA BC + DPSC In hydrogel + MTA or GIC BC + MTA BC + PRF + Biodentine BC + PRP + MTA BC + PRF vs. PRP + Collagen + GIC BC + Synoss putty + MTA BC + Collagen + Portland + MTA BC + LPRF + Portland cement	21 days– 78 months	[66,67,69,73,80,82,83,87,90,91,94,96,97,99,100,101,102,104,105,108,110,112,119,122,124,127,128,130,132]
	Root thickness	BC + hydrogel with FGF+ MTA BC + DPSC In hydrogel + MTA or GIC BC + MTA BC + PRF + Biodentine BC + PRP + MTA BC + PRF vs. PRP + Collagen + GIC BC + Synoss putty + MTA BC + Collagen + Portland + MTA BC + LPRF + Portland cement	21 days– 60 months	[65,66,67,71,80,83,87,90,91,96,97,99,100,101,102,104,105,108,111,112,118,119,122,124,127,128,130,132]
	Apical closure	BC + hydrogel with FGF+ MTA BC + DPSC In hydrogel + MTA or GIC BC + Collagen + coltosol BC + MTA BC + PRF + Biodentine BC + PRP + MTA BC + PRF vs. PRP + Collagen + GIC BC + Synoss putty + MTA BC + Collagen + Portland + MTA BC + LPRF + Portland cement	21 days– 78 months	[8,67,69,70,71,72,73,74,76,77,78,84,88,92,97,98,99,100,102,104,105,110,111,112,118,119,120,122,124,128,132]
	Radiolucy	BC + PRF + MTA BC + MTA or CEM	6 months– 78 months	[8,73,97,102,103,107,108,109,110,119]
	Bone density	BC + hydrogel with FGF+ MTA BC + DPSC in hydrogel + MTA or GIC BC + MTA BC + PRP + MTA BC + Synoss putty + MTA	24–70 months	[70,96,97,109,111,112,120]
	Resorption	BC + PRF + Collagen + Biodentine Medication on different appointment	21 days– 30 months	[117,128,133]
	Calcification in the pulp	BC + MTA or CEM BC + Collagen + MTA BC + PRP + MTA BC + iPRF + Biodentine BC + Amelogen Plus	12–60 months	[66,73,78,85,88,93,96,99,110,118,119,127,133]
	Ligament repair	BC + PRF or PRP + MTA Vs BC + MTA BC + PRP + MTA BC + PRF + Collagen + Biodentine	50 months	[91,95,117]
qPCR	Quantify bacteria	Different appointment medication TAP vs. calcium hydroxide medication	21 days– 19 months	[128,129]
	Cells identification in the canal	Intracanal blood sample after BC	1 month	[135]
Histology	Regenerate tissue observation	BC + MTA BC + Synoss Putty BC + Amelogen Plus BC + Collagen / MTA	7.5–36 months	[106,107,121,127,136]

### 3.4. Risk of Bias

For animal studies, a low risk of selection bias (baseline characteristics) was found. In all studies, the performance bias was unclear, because no information about random housing was given [16,17,18,19,20,21,22,23,24,26,27,28,29,30,31,32,33,34,35,36,37,38,39,40,41,42,43,44,45,46,47,48,49,50,51,52,53,54,55,56,57,58,59,60,61,62,63,64]. Random outcome assessment was scored as low risk for 52% of the studies and unclear for the rest of them. In none of the studies was blinding described, and the risk was rated as unclear. Low risk of attrition and reporting bias was estimated for all studies (Figure 3 and Figure 4).

For randomized clinical studies, randomization was scored as low risk for 65% and medium for the rest. Deviation from the intended intervention was scored as low risk for 55% and the rest was medium. A low risk of missing outcome data was estimated for all studies. Measurement of the outcome was scored as low risk for 25%, while 70% showed medium risk and 5% showed high risk. Moreover, 95% of the studies present a low risk for selection of the reported results. In addition, 40% of studies presented a low risk of bias, 55% medium risk and 5% presented a high risk of bias (Figure 5 and Figure 6).

For human non-randomized studies, all studies presented a low risk for confounding, classification of the intervention, deviation from the intended intention, missing data, and selection of the reported results. However, for the selection of participants, medium risk was noted for all of them. (Figure 7 and Figure 8)

For case reports studies, all the studies presented a low risk of bias (Table 4). In addition, for observational studies, 17 studies presented a low risk, but one study present a mild risk of bias (Table 5).

## 4. Discussion

### 4.1. Success Criteria Assessment of REP

REP has been the subject of numerous studies, both in animals and humans. Its objective is to regenerate intra-canal dentin-pulp tissue that is able to promote root growth in terms of length and thickness and apical closure, while restoring the sensitivity of the tooth and leading to periapical tissue healing [23,138,139,140]. To achieve all these objectives, the ideal protocol has not yet been defined, and there is no clear consensus regarding the scaffolds, disinfection methods, and sealing materials that emerged from the different studies. As research continues, the need for precise, repetitive parameters for assessing success is of prime importance. This systematic review sought to elucidate the different criteria of evaluation in both animals and humans, associated with the risk of bias in the studies.

### 4.2. Ectopic Model

Our research identified that several rodent ectopic models of REP, involving subcutaneous implantation of DPSCs and/or growth factors such as VEGF into the tooth slice or tooth root, have been developed to assess the biocompatibility and regenerative potential of biomaterials with a follow-up time of 12 days to 3 months [16,17,18,19,20,21]. 

The histological criteria for success in these models are the regeneration of pulp-like tissue with the presence of the odontoblastic palisade and neo-vessels along the root or the tooth edge [16,17,20]. Regenerated tissue should fill the entire root space. Often, calcifications can be detected if cement tissue is present in the apical region or if pulp-like tissue is formed in the middle third. Furthermore, the presence of macrophages along the vessels to control the inflammation was assessed [77]. 

In this review, pulp-like tissue was obtained in almost all ectopic studies. The observations made in histological sections were essentially based on the presence of vascularization at different levels of the root or tooth slice. The presence of DSPP labelling was also found very often, which can confirm the presence of odontoblast-like cells. In a rabbit study, there was the presence of osteodentine-like tissue. Some authors showed that the formation of an atubular fibrodentine or osteodentine matrix is a precursor to the formation of a more organized tubular matrix [141,142,143].

Obviously, ectopic models represent a primary step in research focusing on REP. On the other hand, thee subcutaneous area sensibly differs from the oral cavity and periapical region mostly in terms of blood supply. Indeed, the vascularization of the subcutaneous tissue is very different from the vascularization of the normal pulp in the intra-canal area [29]. It has also been demonstrated that the transplantation of long dental fragments with a relatively closed apex (less than one millimeter) leads to inconsistent results due to the difficulty for the vascular and nervous system of the mouse to reach the interior part of the dental fragment [20,144,145]. Moreover, ectopic studies are also not very reliable due to the mixed cell population, with both animal and human regenerated tissue obtained. In addition to all these drawbacks, the technique used for REP is very different from those performed on humans, because the authors strongly suggest that, after the results are obtained by the ectopic models, tests on an orthoscopic model be performed [146].

### 4.3. Animal Models

Regarding small-animal models of REP, only rats and ferrets were involved. The procedure was performed on the incisors, which are continuously growing teeth, while in ferrets, REP was performed on canine teeth. The limited size of small animals and the substantial anatomical differences can impact the complete removal of the pulp and, therefore, the regeneration process [26].

Excluding studies on sheep and mini-pigs using mono-rooted or continuously growing teeth, the molar-premolar were the most used teeth. Multirooted teeth are good models allowing for the reproduction of most common clinical tables, such as deep caries inducing pulp necrosis [147].

The most used biomaterials were trisilicate cement, such as MTA in association with hydrogels seeded or not with DPSC, PRF or PRP, or a blood clot.

Indeed, DPSCs are the precursors of odontoblast-like cells [148] and, as previously demonstrated, are capable of regenerating the pulp–dentin complex in vivo [143]. These cells are often of human origin, harvested from third molars or teeth extracted for orthodontic reasons. Moreover, their association with hydrogel allows one to keep the cells in a matrix that will disintegrate with time [35,97]. PRP and PRF are widely used in regenerative dentistry, since they are known to optimize healing pathways by stimulating the scar stem cells present in the injured area.

The use of a blood clot allows the formation of even more entanglement for optimal regeneration. Thus, the recovery and regeneration of the pulp structure and function of the pulp tissues were achieved. In addition, the follow-up was long enough to observe a complete formation of the tissue and an increase in the root walls for small animals as well as large animals.

However, these procedures requiring cell cultures or blood derivates are hardly reproducible in common clinical practice [147,149].

The histological success criteria in animals were the resolution of the apical lesion, the presence of vital pulp-like tissue, the formation of mineralized neo-tissue, the closure of the apex with the possibility of newly formed blood vessels, and the presence of nerve fibers. 

It can be observed that histological studies were almost systematically performed by distinguishing three regions along the root and characterizing the tissue. 

The coronal third was very often at an early stage of hard tissue development on the canal walls. It is assumed that the migrating stem cells differentiate into cementoblast-like cells and deposit a matrix of collagen fibers. These fibers calcify, forming cementum islands. The further down the canal, the more these islands will fuse and form a thin layer of cementum in the second region, which was more medial. At the apical level of this region, which is closer to the apical region, a thicker, acellular matrix, such as immature acellular cementum, was found. The pulp cavity was filled with loose fibrovascular tissue. Finally, in the third region, there was mature hard tissue covering the canal walls and loose connective tissue covering the pulp canal. The hard tissue is also cementum. 

Radiologically, the same characteristics were observed: Periapical lesion resolution, an increase in root length and the thickness of the root length, and a decrease in the apical diameter. In all cases, REP allowed for the healing of the periapical lesion, to decrease the diameter of the apex and the lengthening of the roots. 

Inflammation is essential for tissue regeneration. The goal is to have sufficient but controlled inflammation [150]. In many animal studies, the presence of persistent periapical inflammation or intracanal and periapical inflammation may be due to remnants of intracanal medicaments or attempted healing in the newly formed tissue [24,25,50,61]. Closure of the apex is a sign of success that assures that root building is somehow complete [2,23,27,32,41,42,43,46,49,51,57,58,59,66]. However, the degree of mineralization must also be assessed, since dentin-type mineralization is not systematically found in small and large animals. Most of the time, a bone-like or cementum-like mineralized extension at the apex is found. Cement islands are also very often found in the newformed tissue. Despite the use of DPSC, there is no formation of pulp tissue with an intact cell layer similar to an odontoblast. However, in some samples, connective tissue could be observed inside the root canal with cells. The presence of ligament-like newformed tissue is also noticed. Based on large and small animal studies, stem/progenitor cells introduced into the root canal spaces of immature teeth with necrotic pulp after revascularization procedures appear to be able to differentiate into cementoblast- or osteoblast-like cells rather than odontoblasts. These stem/progenitor cells are likely derived from the periodontal ligament or periapical alveolar bone marrow, because the newly formed tissues in the canals of revascularized teeth are cementum or bone-like tissue [50,57,107,149].

Some studies have found pulp-like tissue, but it was often due to the previous presence of healthy pulp tissue [23]. This does not represent the clinical reality where REP treatment is performed in cases of necrotic pulp.

To summarize, in small animal ectopic model studies, regenerated tissues are very similar to the dentine pulp. 

In small animals, REP is performed on healthy pulp. The fact that the tooth is small, resulting in infected pulp, makes the tooth even more fragile, which makes REP difficult to perform. However, some studies have been successful in performing REP on infected pulp. 

In large animals, the pulp is most often infected, which is simpler to perform and closer to the clinic. To assess the inflammation, the use of radiography allows one to observe whether a preapical lesion is present or not. 

### 4.4. Clinical Studies 

In human studies, the primary outcomes were the absence of pain, inflammation, or swelling. The resolution of the periapical lesion was observable in 2D or 3D radiography. 

To assess regeneration, different clinical tests were performed. In this review, a few studies succeeded in having positive responses in sensitivity tests [71,74,80,92,95,96,101,102,106,109,121,122,123,126,129,130,132,133], confirming that neurogenesis is variable in REP [4,11].

Radiographically, almost all the studies reported the resolution of apical lesions, very often associated with high apex closure. Periapical radiographs with a parallel technique also allow one to verify the increase in radicular length, the thickening of dentinal walls, and periapical tissue repair. It is important to consider the normal growth of the patients, as most of them are in their growing phase, which may create visual variations in the measurements [85].

In all studies, the first radiographic finding is complete apical closure with a decrease in the apical diameter, while the send radiographic outcome was thickening of the canal walls with average wall thickening and continued root elongation. 

Moreover, intrapulpal calcifications were often found close to the blood clot used alone, or with scaffold materials. 

When performed, histological examination showed the formation of intracanal mineralized tissue around the scaffold particles solidifying with newly formed cementum tissue along the dentinal walls. Quantitative PCR confirmed the absence of odontoblasts able to make pulp-like tissues. 

Regenerated tissue, whether for REP in animals or humans, is a fibrous pulp-like tissue with cementum-based islands in the middle and mineralization along the dentin walls. Sometimes, cementoblasts are found by immunohistochemistry. One could assume that internal calcification plays a protective role, but the evolution of such calcified tissues in the long term are still not known [92]. 

With the wide use of the REP procedure, several questions have arisen, mostly concerning the viability of the treated teeth. A tooth is considered viable once the root is structured and the tooth is clinically and radiographically asymptomatic with no signs of failure.

In the clinical studies included in this review, success of REP therapy was represented by the persistence of the tooth remaining asymptomatic. Only few patients showed positive responses to pulp sensitivity tests after revitalization. 

Data arising from prolonged follow-up periods are still lacking. In fact, all randomized studies report follow-up periods that do not exceed 2 years. In the case of retrospective studies or case reports, we found two cases in which the follow-up period was between 6 and 8 years. 

### 4.5. Risk of Bias

The included studies presented a low risk of bias in terms of animal selection (ARRIVE guidelines were respected), attrition, and reporting. On the other hand, weak reporting in terms of performance and detection affected evaluations and the synthesis of results. Thus, SYRCLE guidelines should be followed, especially for randomization protocols, animal housing facilities, and blinding, which could improve homogeneity of small and large animal model trials focusing on REP. 

Regarding randomized studies, in only 40% of the cases was there an overall low risk of bias. Indeed, there was a lack of information in terms of randomization, deviation from the intended intervention, and measurement of the outcome, complicating comparisons between studies. 

For non-randomized studies, a medium risk of bias in patient selection was found.

Indeed, more structured protocols for patient enrollment should be applied.

An overall low risk of bias was found in retrospective and case reports studies, since the criteria of selection, ascertainment, causality, and reporting were respected.

## 5. Conclusions

Currently, regenerative endodontics is legitimately considered part of the spectrum of endodontic therapies. In fact, REP on immature or mature teeth is a reliable approach that creates a new tissue, ensuring tooth development and viability.

However, no study has succeeded in regenerating pulp-like tissue; instead, in both preclinical and clinical studies, ligamentous tissue with cementum or bone-like mineralization replaced the necrotic pulp. Intra pulpal calcification can play a protective role for the regenerated tissue, but the evolution of this calcification in the long term is not known. 

Some animal studies reported a vascularized and innervated regenerated pulp, but the response to the clinical test was not verified. In human studies, only few patients have regained sensitivity after revitalization.

Preclinical and clinical studies identify the success of REP therapy as the persistence of the tooth without signs or symptoms of failure. 

Even if endodontic reparation can clinically satisfy the needs of dental and alveolar bone development and preservation, further studies are still necessary to identify procedures to successfully reproduce the physiological structure and function of the dentin–pulp complex. 

## Figures and Tables

**Figure 1 ijms-23-10534-f001:**
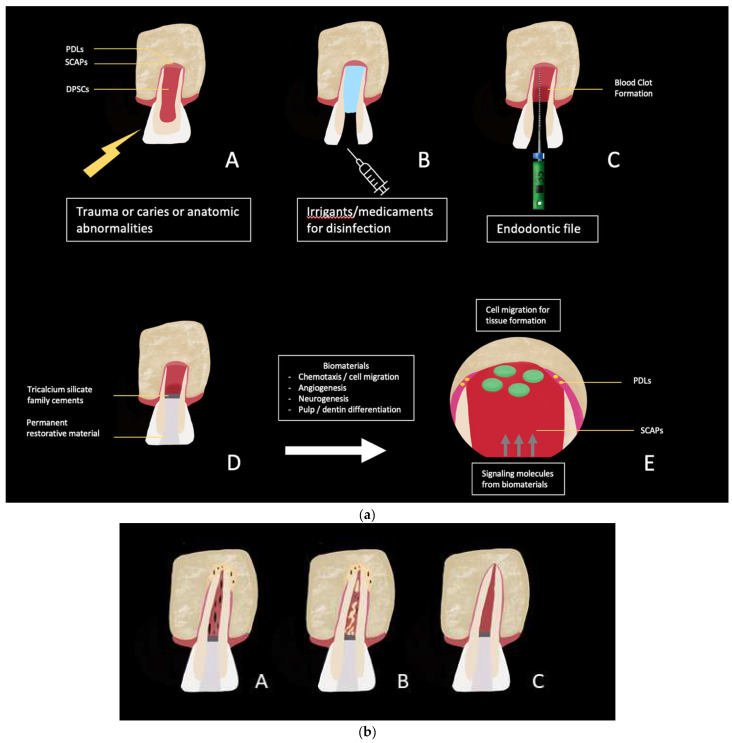
(**a**) Schema illustrating REP procedure. (**A**) Immature permanent incisor tooth exposed to trauma or caries. (**B**) Access cavity preparation and chemical debridement by using of irrigants and/or medicaments. (**C**) Bleeding induced by dental endodontic File to create a blood clot. (**D**) Restoration of the access cavity with a permanent restorative material covering the biomaterial in contact with blood clot or with collagen plug, PRF, PRP, etc. (**E**) Release of signaling molecules from biomaterial (growth factors, calcium silicate, depending on the material). They influence SCAPs (from apical papilla) and PDLs, including chemotaxis/cell migration, angiogenesis, neurogenesis, and differentiation into pulp/dentin complex. DPSCs (Dental Pulp Stem Cells); SCAPs (Stem Cells from Apical Papilla); PDLs (Periodontal Ligament cells) (Document of URP2496). (**b**) Different type of achievement of tissue regeneration determined by continued root development, increased dentinal wall thickness by cementum-like deposition, and apical closure. (**A**) Immature permanent incisor tooth after REP technique with ”pulp-like” tissue with ligamentous, fibrous aspect and apical closure with mineralized tissue: Cement-like or bone-like tissue; (**B**) ”pulp-like” tissue with ligamentous aspect and apical closure with mineralized tissue: Cement-like or bone-like tissue. Cementum island in the intracanal tissue; (**C**) the expectation of the pulp-like tissue with an increased root thickness and length, a decreasing apex diameter. Document of URP2496.

**Figure 2 ijms-23-10534-f002:**
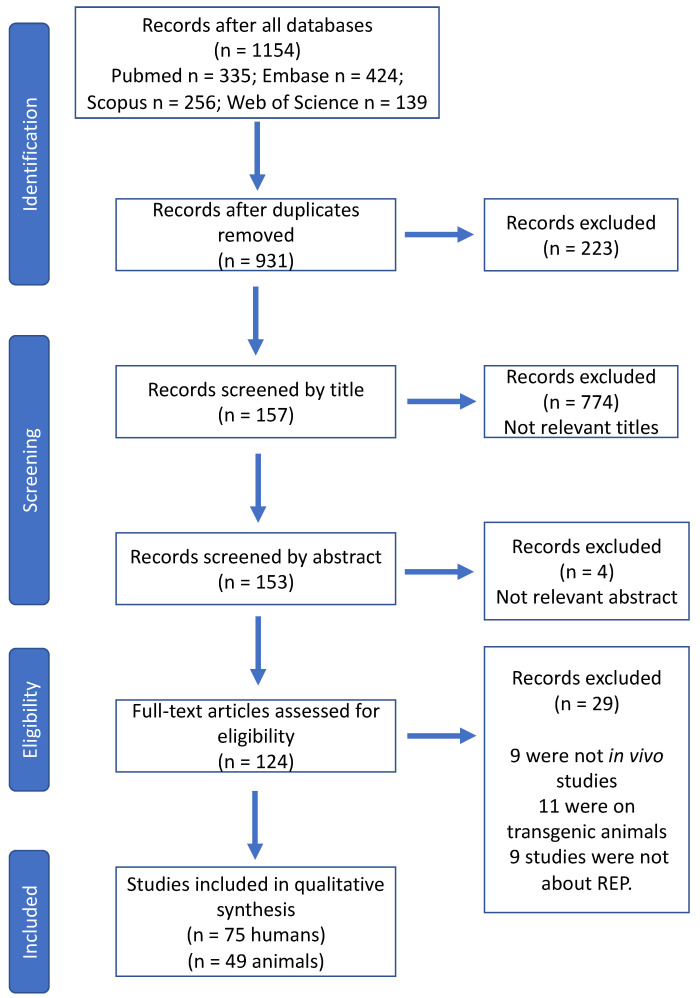
Flowchart of the article selection process.

**Figure 3 ijms-23-10534-f003:**
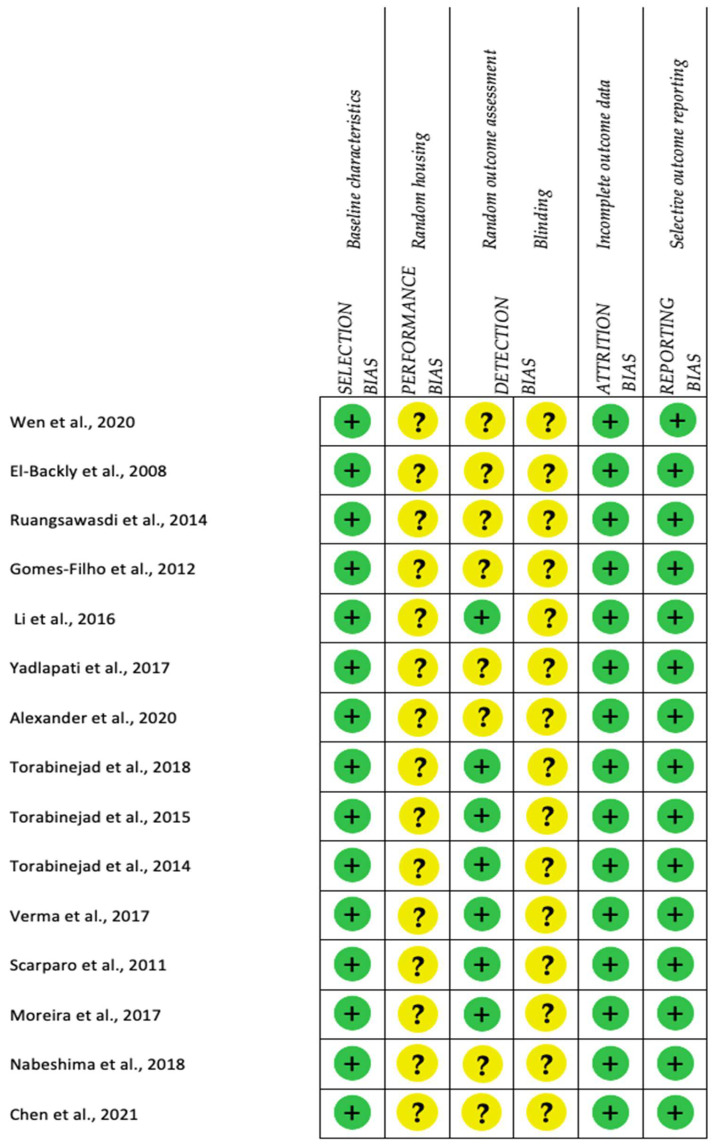
Risk of bias assessment of REP in animal studies according to the Systematic Review Centre for Laboratory Animal Experimentation (SYRCLE): Authors’ judgment about each risk of bias item (green = low, yellow = unclear) [16,17,18,19,20,21,22,23,24,25,26,27,28,29,30,31,32,33,34,37,38,39,40,41,42,43,44,45,46,47,48,49,50,51,52,53,54,55,56,57,58,59,61,62,63,64,65].

**Figure 4 ijms-23-10534-f004:**
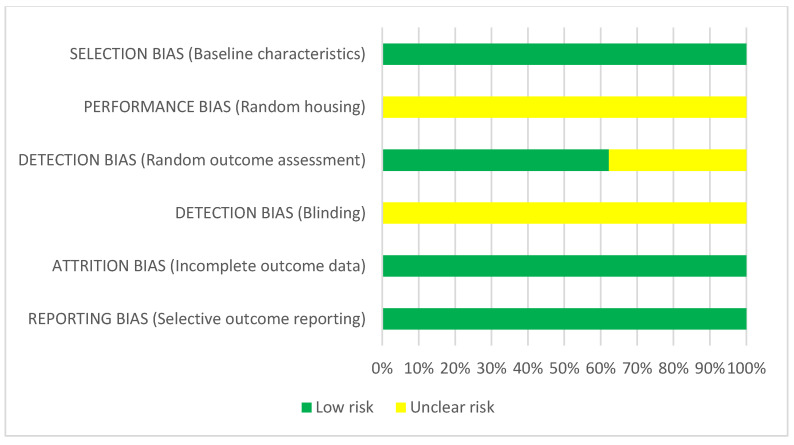
Risk of bias assessment of REP in animal studies according to the Systematic Review Centre for Laboratory Animal Experimentation (SYRCLE) (green = low risk, yellow = unclear risk).

**Figure 5 ijms-23-10534-f005:**
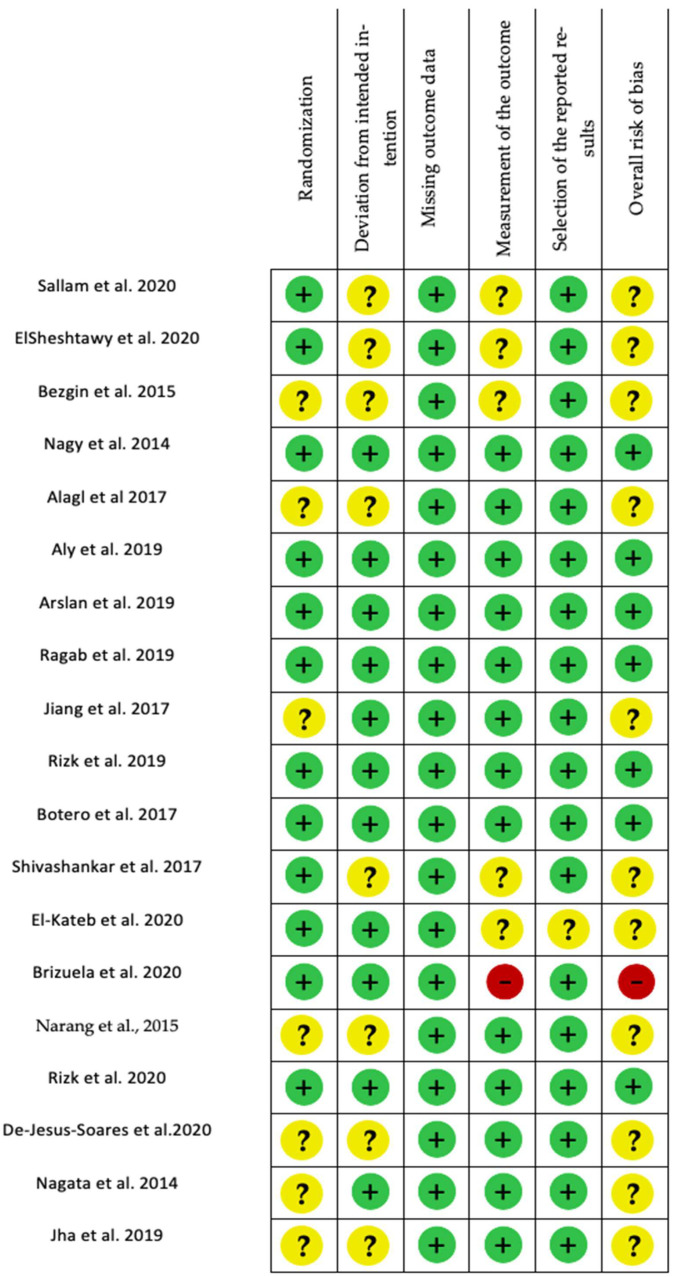
Risk of bias assessment evaluated according to the R.O.B 2.0. Authors’ judgment about the following items: Randomization, deviation from intended intervention, missing outcome data, measurement of the outcome, selection of the reported results, and overall risk of bias (green = low, yellow = unclear, red = high). [6,7,69,70,84,85,86,87,88,89,90,114,115,123,126,128,129,135].

**Figure 6 ijms-23-10534-f006:**
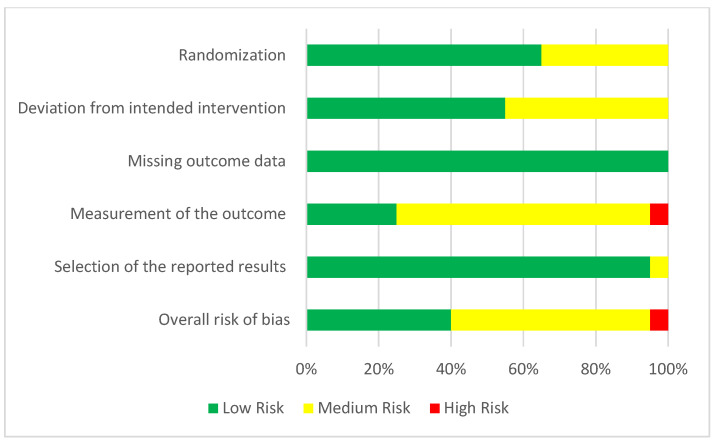
Risk of bias assessment evaluated according to the R.O.B 2.0 tool. Randomization, deviation from intended intervention, missing outcome data, measurement of the outcome, selection of the reported results, and overall risk of bias (green = low, yellow = moderate, red = high).

**Figure 7 ijms-23-10534-f007:**
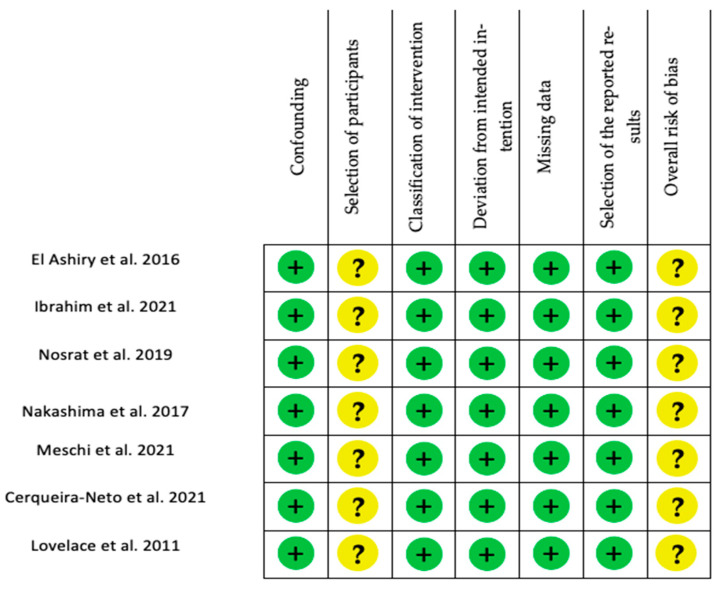
Risk of bias summary: Authors’ judgement about each risk of bias item for each included non-randomized study (Robins I tool). [60,116,121,123,124,132,135].

**Figure 8 ijms-23-10534-f008:**
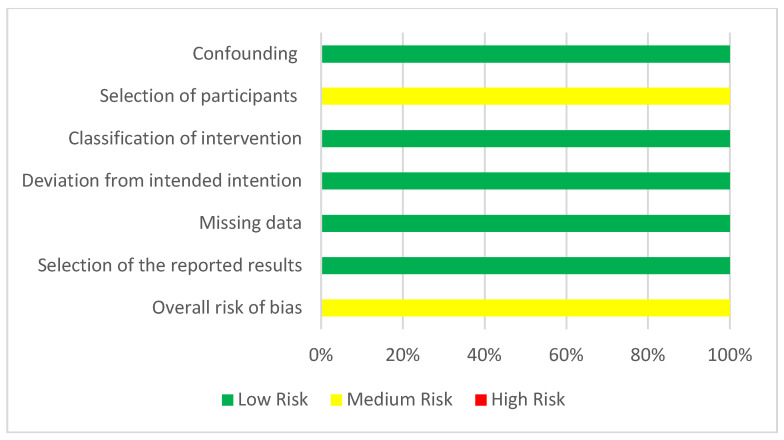
Risk of bias assessment evaluated according to the ROBINS I tool. Confounding, selection of participants, classification of intervention, deviation from intended intervention, missing data, selection of the reported results, overall risk of bias (green = low, yellow = moderate, red = high).

**Table 1 ijms-23-10534-t001:** Ectopic REP techniques in animal studies.

Animal Models: Ectopic Procedure				
Assessment	Main Results	Procedure	Follow-Up	Model
Histology	Soft tissue formation	Root Tooth: VEGF + DPSCs + MTA Dentin slice: rBMSC + collagen scaffold + iRoot BP Human teeth roots + fibrin gel DPSCs + polymers scaffold	12 days–3 months	Mice [20] Rats [16,18] Rabbit [17]
	Presence of odontoblasts cells	Root Tooth: VEGF + DPSCs + MTA Dentin slice: rBMSC + collagen scaffold + iRoot BP Human teeth roots + fibrin gel DPSCs + polymers scaffold	12 days–3 months	Mice [20] Rats [16,18] Rabbit [17]
	Presence of Inflammation	Polyethylene tubes: TAP vs. CHP calcium VEGF-loaded fiber + root fragment + MTA	1.5–3 months	Mice [21] Rats [19]
	Presence of Vessels	Polyethylene tubes: TAP vs. CHP calcium Dentin slice: rBMSC + collagen scaffold + iRoot BP Human teeth roots + fibrin gel DPSCs + polymers scaffold	12 days–3 months	Mice [20,21] Rats [16,18] Rabbit [17]
	Presence of Nerves	Dentin slice: rBMSC + collagen scaffold + iRoot BP Root Tooth: VEGF + DPSCs + MTA	2–3 months	Mice [20] Rats [16]
	Presence of mineralization	Polyethylene tubes: TAP vs. CHP calcium	3 months	Rats [19]

**Table 4 ijms-23-10534-t004:** Risk of bias assessment of case reports according to an adapted Newcastle–Ottawa Scale with the following criteria: Selection, ascertainment, causality, and reporting.

Author/Year	Selection	Ascertainment		Causality	Reporting	Results	Finality
Yoshpe et al., 2021 [8]	1	1	1	1	1	5	Low
Jiang et al., 2020 [93]	1	1	1	1	1	5	Low
Sabeti et al., 2021 [94]	1	1	1	1	1	5	Low
Gaviño et al., 2017 [95]	1	1	1	1	1	5	Low
Terauchi et al., 2021 [96]	1	1	1	1	1	5	Low
Jung et al., 2008 [97]	1	1	1	1	1	5	Low
McTigue et al., 2013 [67]	1	1	1	1	1	5	Low
Li et al., 2017 [99]	1	1	1	1	1	5	Low
Saoud et al., 2014 [100]	1	1	1	1	1	5	Low
Dabbagh et al., 2012 [101]	1	1	1	1	1	5	Low
Dudeja et al., 2015 [102]	1	1	1	1	1	5	Low
Ulusoy et al., 2017 [103]	1	1	1	1	1	5	Low
Cehreli et al., 2011 [104]	1	1	1	1	1	5	Low
Sachdeva et al., 2015 [105]	1	1	1	1	1	5	Low
Lin et al., 2014 [106]	1	1	1	1	1	5	Low
Becerra et al., 2014 [107]	1	1	1	1	1	5	Low
Chen et al., 2013 [108]	1	1	1	1	1	5	Low
Chang et al., 2013 [109]	1	1	1	1	1	5	Low
Lenzi et al., 2012 [110]	1	1	1	1	1	5	Low
Shin et al., 2009 [111]	1	1	1	1	1	5	Low
Shiehzadeh et al., 2014 [112]	1	1	1	1	1	5	Low
Plascencia et al., 2016 [113]	1	1	1	1	1	5	Low
Yoshpe et al., 2020 [117]	1	1	1	1	1	5	Low
Bakhtian et al., 2017 [118]	1	1	1	1	1	5	Low
Mehrvarzfar et al., 2017 [119]	1	1	1	1	1	5	Low
Cymerman et al., 2020 [120]	1	1	1	1	1	5	Low
Shimizu et al., 2013 [127]	1	1	1	1	1	5	Low
Nazzal et al., 2018 [130]	1	1	1	1	1	5	Low
Meschi et al., 2016 [136]	1	1	1	1	1	5	Low

**Table 5 ijms-23-10534-t005:** Risk of bias assessment of observational studies (cohort studies) according to the Newcastle–Ottawa Scale with the following criteria: Selection and outcome.

Author/Year	Selection			Outcome			Results	Finality
Meschi et al., 2018 [72]	1	1	1	1	1	1	6	Low
Elfrink et al., 2021 [73]	1	1	1	1	1	1	6	Low
Pereira et al., 2020 [74]	1	1	1	1	1	1	6	Low
Chrepa et al., 2020 [75]	1	1	1	1	1	1	6	Low
Mittman et al., 2020 [76]	1	1	1	1	1	1	6	Low
Linsuwanont et al., 2017 [77]	1	1	1	1	1	1	6	Low
Estefan et al., 2016 [78]	1	1	1	1	1	1	6	Low
Peng et al., 2017 [79]	1	1	1	1	1	1	6	Low
Chen et al., 2016 [80]	1	1	1	1	1	1	6	Low
Jeeruphan et al., 2012 [81]	1	1	1	1	1	1	6	Low
Bukhari et al., 2016 [82]	1	1	1	1	1	1	6	Low
Chan et al., 2017 [83]	1	1	1	1	1	1	6	Low
Song et al., 2017 [84]	1	1	1	1	1	1	6	Low
Zizka et al., 2021 [98]	1	1	1	1	1	1	6	Low
Meschi et al., 2019 [125]	1	1	1	1	1	1	6	Low
Bose et al., 2009 [131]	1	1	1	1	1	1	6	Low
Shah et al., 2012 [137]	1	1	1	1	1	1	6	Low
Sutam et al., 2018 [133]	1	1	1	1	0	0	4	Mild

## Data Availability

The datasets generated for this study can be obtained upon reasonable request by email to the corresponding author.

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
