# Peer review of "Tissue Characteristics in Endodontic Regeneration: A Systematic Review"

_ijms, 2022, doi:10.3390/ijms231810534_

Round 1
Reviewer 1 Report
Excellent efforts, and here are my notes:
Title: The title use reparative wording, I may for suggest the following
Tissue characteristics in endodontic regeneration: A systematic review
Or maybe
Success Evaluation Criteria in Endodontic Regeneration
1- Introduction:
Paragraph from line 47 to 61 is long and need more references. The last statement in that paragraph also need a reference.
Unclear rationale for the review, what is already known was not clearly mentioned in a smooth flow.
The aim at the end of the paragraph: “Thus, this systematic review aimed to bring out criteria of successful pulp-complex regeneration procedure, by analyzing the different assessment methods used, both in animals and in humans’ studies.”
This aim doesn’t seem related to the statement of the title, which is “characteristics of regenerated tissues”
Fig 1 need reference or clarification of the source. Is it prepared by the Author?
The author used the PRISMA systematic review protocol was used, but I have to say that there was no explicit statement of the question(s) in the review that should address with reference to participants, interventions, comparators, and outcomes (PICO)
2- Results:
I'm not quite sure if this is the right subtitle for this section, as the Materials and Methods has not been explained yet. The Method was explained at the end of the paper before the conclusion starting Line 510
Line 91 “130 articles met the inclusion criteria”: I didn’t really find clear inclusion criteria. The author need to specify the characteristics beside the PICO. Things like the study design, setting, time frame, characteristics like years considered, language, publication status, to be used as criteria for eligibility for the review early in the paper?
The process used for selecting studies and data extraction (One author? Two independent reviewers?). The details are not clear in the flowchart, with no clear reasoning for exclusion.
How the outcome was assessed? How the best success criteria was determined by the author?
The results presentation by including the selected paper summary is great but I'm not sure if it can be shortened as it seems very long, in the PDF from page 5-72 included all the papers summaries which were then summarized with combined results in smaller tables and /or graphs.
Conclusion:
“REP technique on immature or mature teeth is a more reliable procedure than apexification since, by creating a new tissue, it assures tooth development and viability.”
I'm not sure if the previous statement was the core of the study and if it was supported by enough evidence and properly carried out analysis to prove?
- Any clinical relevance ?
- The cited references are relatively recent relevant
Author Response
Good Morning,
Please, see the attachment.
Thank you.
The authors.

Reviewer 2 Report
Dear Authors!
Congratulations for the big work you have completed. Please allow me to raise a few issues and questions:
In line 5 of the abstract, there is a missing space between Abstract 5. “by REP.A customize” Please add a space after then end of the sentence.
In the abstract, in line 22, the authors write “REP procedure”, however, earlier they state REP is “Regenerative endodontic procedure” This means, you shall not have the word procedure after the REP, it already contains.
In line 38, the authors state: “recommended by several endodontic and paediatric dentistry associations”, but there are no references. Please add relevant references, where you can find the recommendations be the above mentioned associations.
In Figure 1. There are some abbreviations, which are not explained anywhere. Though some of them are quite obvious, please explain exactly all abbreviations either in the text section, or even in the figure caption part.
There is a discrepancy between the text and Figure 2. In the text, it is written, that 130 articles eligible, 75 were clinical studies and 45 animal ones. This itself does not seem to be correct, as 75+45 add together only 120 not 130. Also in Figure 2, the relevant numbers are 123, 75 and 48. Please explain, what are the correct numbers, and why we do have this discrepancies? Also please make sure, you have the right numbers in the abstract as well.
Moreover these, on Table 1. We can see 49 animal model articles. How is this possible? The same 49 articles can be seen in Figure 3. Please explain.
The display of Table 1. Is very disturbing. It lasts for too many pages, please make sure, you discuss this with the editor and change the format to a readable one.
In Figure 6. it is not clear, if it reflecting still on the above mentioned animal studies or on all articles? All other figures above contain the data, that they reflect on the animal studies, this one does not.
In the discussion section, there are weird sentences, starting with small letter, not capital letter, like line 472 or 486. Please correct the language editing precisely.
For me it is not really ok, to have the methods section after the discussion. Why is it placed there? It shall be before results, after introduction. Please change or explain, why it shall be better after discussion?
Author Response
Good Morning,
Please, see the attachement.
Thank you
The authors.

Round 2
Reviewer 1 Report
Thank you.
Fine-tuning will be helpful.